# GLUT3/SLC2A3 Is an Endogenous Marker of Hypoxia in Prostate Cancer Cell Lines and Patient-Derived Xenograft Tumors

**DOI:** 10.3390/diagnostics12030676

**Published:** 2022-03-10

**Authors:** John M. Ryniawec, Matthew R. Coope, Emily Loertscher, Vignesh Bageerathan, Diogo de Oliveira Pessoa, Noel A. Warfel, Anne E. Cress, Megha Padi, Gregory C. Rogers

**Affiliations:** 1Department of Cellular and Molecular Medicine, University of Arizona Cancer Center, University of Arizona, Tucson, AZ 85719, USA; ryniawecj@arizona.edu (J.M.R.); mcoope@arizona.edu (M.R.C.); eloerts2@email.arizona.edu (E.L.); warfelna@arizona.edu (N.A.W.); 2Biostatistics and Bioinformatics Shared Resource, University of Arizona Cancer Center, University of Arizona, Tucson, AZ 85724, USA; vbageerathan@arizona.edu (V.B.); dpessoa@arizona.edu (D.d.O.P.); 3Department of Molecular and Cellular Biology, University of Arizona Cancer Center, University of Arizona, Tucson, AZ 85721, USA

**Keywords:** hypoxia, prostate cancer, glucose transporter, SLC2A3, GLUT3

## Abstract

The microenvironment of solid tumors is dynamic and frequently contains pockets of low oxygen levels (hypoxia) surrounded by oxygenated tissue. Indeed, a compromised vasculature is a hallmark of the tumor microenvironment, creating both spatial gradients and temporal variability in oxygen availability. Notably, hypoxia associates with increased metastasis and poor survival in patients. Therefore, to aid therapeutic decisions and better understand hypoxia’s role in cancer progression, it is critical to identify endogenous biomarkers of hypoxia to spatially phenotype oncogenic lesions in human tissue, whether precancerous, benign, or malignant. Here, we characterize the glucose transporter GLUT3/SLC2A3 as a biomarker of hypoxic prostate epithelial cells and prostate tumors. Transcriptomic analyses of non-tumorigenic, immortalized prostate epithelial cells revealed a highly significant increase in GLUT3 expression under hypoxia. Additionally, GLUT3 protein increased 2.4-fold in cultured hypoxic prostate cell lines and was upregulated within hypoxic regions of xenograft tumors, including two patient-derived xenografts (PDX). Finally, GLUT3 out-performs other established hypoxia markers; GLUT3 staining in PDX specimens detects 2.6–8.3 times more tumor area compared to a mixture of GLUT1 and CA9 antibodies. Therefore, given the heterogeneous nature of tumors, we propose adding GLUT3 to immunostaining panels when trying to detect hypoxic regions in prostate samples.

## 1. Introduction

Oxygen consumption rates are high in tumors due to the proliferative nature of cancer cells and to the presence of infiltrating immune cells. Consequently, uncontrolled proliferation of cancer cells causes tumors to outgrow their blood (and, thus oxygen) supply, leading to regions with low oxygen levels, known as hypoxia [1]. Although solid tumors can compensate by promoting angiogenesis, the new vasculature is often poorly constructed with leaky vessels, blind ends, and shunts [2]. Fluctuations in interstitial pressure, coupled with the absence of lymphatic drainage, can cause sporadic vascular collapse, resulting in cyclic (or intermittent) hypoxia, often heterogeneously distributed within the tumor [3,4]. Although early estimates suggested that 50–60% of solid tumors contain regions of hypoxia or anoxia (a lack of oxygen) [5], it is now well established that hypoxia is a hallmark of most, if not all, solid tumors [4,6].

Hypoxia has a profound impact on tumorigenesis and cancer progression because low oxygen tension selects for stress-tolerant tumor cells with enhanced genomic instability, defects in apoptosis, and an aggressive cancer phenotype [7,8,9,10]. Cells exposed to acute hypoxia (low intracellular oxygen levels that last minutes to hours) or chronic hypoxia (lasting hours to days) exhibit dramatic changes in gene expression due to the stabilization of hypoxia-inducible factors (HIF), HIF-1α and HIF-2α These HIF transcription factors are master-regulators of the hypoxia response, accounting for the bulk of transcriptional changes in hypoxia [11,12]. Hypoxia and activation of the HIF-1α/2α gene networks participate in numerous cancerous behaviors including cell immortalization, most aspects of the metastatic cascade, vascularization, and suppression of the immune response [11,13,14,15,16,17]. Not surprisingly, low oxygen levels are associated with resistance to chemotherapy and radiotherapy, as well as poor patient survival [18,19,20,21].

In hypoxia, a reduced efficiency of electron transfer in mitochondria generates an excess of reactive oxygen species (ROS) [22]. To suppress ROS and meet their high energy demands, proliferating tumor cells adapt to oxygen deprivation by switching from aerobic to anaerobic metabolism, a hallmark of solid tumors known as the Warburg effect [23,24]. Some direct targets of HIF transcription facilitate anaerobic glycolysis by increasing glucose uptake or combatting tumor acidification. A common HIF target, carbonic anhydrase IX (CA9), is a membrane-bound enzyme that regulates both extracellular and intracellular compartmental pH [25,26,27]. In general, CA9 expression is suppressed in normoxia, but highly upregulated by hypoxia in a variety of cancer cell types [28,29]. Moreover, CA9 has been localized to perinecrotic areas within tumors and associates with a poor patient prognosis [30,31,32,33,34,35]. Another well-established HIF-1α target that contributes to a tumor cell’s high rate of glucose uptake is the glucose transporter 1, GLUT1 (encoded by the SLC2A1 gene) [36,37,38,39,40]. GLUT1 is ubiquitously expressed and also upregulated by hypoxia in several in vitro and in vivo tumor models, where it contributes significantly to tumor growth [1,41].

Hypoxia is especially pathologically relevant in prostate cancer because it is associated with biochemical relapse after radiotherapy and local recurrence in the prostate gland, induces androgen-independent growth, and promotes disease progression [42,43,44,45]. Physiological oxygen levels (physoxia) are normally quite low in the prostate (>5% median oxygen) compared to other tissues, and can plummet to 0.3% median oxygen in prostate cancer [3]. Given the influence of hypoxia on malignant transformation, detection of hypoxia in normal tissue and tumors is an important goal which could aid in personalized therapeutic decisions. One strategy is to utilize immunohistochemistry or fluorescence microscopy to spatially phenotype hypoxia within tissues. In this regard, CA9 and GLUT1 have been examined as endogenous biomarkers of hypoxia, specifically chronic hypoxia, because low oxygen levels increase their expression [10]. However, in the case of CA9, its expression in prostate cancer is inconsistent; CA9 is highly expressed in androgen-independent prostate cancer cell lines in hypoxia, but absent or moderate in primary prostate cancer and benign prostatic hypertrophy (BPH) [26,46,47,48,49]. GLUT1 is expressed in cultured prostate cancer LNCaP cells as well as BPH but absent in malignant prostate tumors [50]. Thus, there is a need to explore new hypoxia biomarkers in prostate tumors while considering that the best option may be to probe for a panel of intrinsic markers [51]. Here, we demonstrate that the glucose transporter, GLUT3, is highly expressed in non-tumorigenic prostate epithelial cells and prostate cancer cells when cultured in hypoxia. Coupled with its co-localization with hypoxia markers in mouse xenograft tumors and PDX prostate cancer models, our findings suggest that GLUT3 should be included in a panel of endogenous biomarkers of hypoxia for further clinical investigation.

## 2. Materials and Methods

### 2.1. Cell Culture

DU145 cells were obtained from the American Type Culture Collection (ATCC, #HTB-81). CTN-9 cells were derived from immortalized human PrEC cells and transformed by transient centrosome removal which induced chromosomal instability [52]. CTN1-2 and CTN2-2 lines were isolated from mouse xenograft tumors formed from CTN-9 cells [52]. CTN-9, CTN1-2, CTN2-2, and DU145 cells were cultured in Iscove’s modified Dulbecco’s medium (IMDM; Corning, Manassas, VA, USA, cat. no. 10-016-CV), supplemented with 10% fetal bovine serum containing detectable levels of testosterone and corticosteroids (FBS, Omega Scientific, Tarzana, CA, USA, FB-11), and 100 IU penicillin, 100 μg/mL streptomycin (Corning, Manassas, VA, USA, Product No. 30-002-CI). Immortalized RWPE-1 cells (ATCC, CRL11609TM) were cultured in Keratinocyte Serum Free Medium (K-SFM; ThermoFisher, Grand Island, NY, USA, 10724-011) with 0.05 mg/mL BPE, 5 ng/mL EGF. All cells maintained at 37 °C in a 5% CO_2_ atmosphere. Hypoxia experiments were performed at 1% O_2_ using a H35 Hypoxystation (Don Whitley Scientific, West Yorkshire, UK).

### 2.2. RNA-Seq Sample Preparation

Triplicate samples of cultured RWPE1 cells were prepared in normoxia (20% O_2_) and cells shifted to hypoxia (1% O_2_) for 6, 12, and 24 h. RNA was isolated using the Qiagen RNeasy mini kit (Qiagen, Hilden, Germany, catalog #74104), which uses size exclusion to isolate all RNAs longer than 200 nucleotides. Homogenization of RWPE1 cells was performed using a 22-gauge hypodermic needle as instructed by the RNeasy protocol. Genomic DNA was removed from samples using Qiagen RNase-Free DNase Set (Qiagen, Hilden, Germany, catalog #79254) as instructed in the RNeasy protocol. RNA was sequenced at the University of Arizona Genomics Core.

### 2.3. RNA-Seq Data Processing

Raw reads were trimmed with Trimmomatic Version 0.39 in single end mode with a phred score threshold of 33, using the universal TruSeq-3-SE adapter reference and retaining reads at least 36 base long. Processed reads were aligned with STAR using a GRCh38 reference genome (version 99). Gene level counts were generated with featureCounts from the Rsubread package. Low-expressing genes were removed using filterByExpr with default parameters from edgeR. Counts were normalized using TMM and voom from edgeR, with the time points (0 h, 6 h, 12 h, and 24 h) being the experimental conditions specified in the design matrix.

Principal Component Analysis (PCA) was performed using prcomp in R, and differential expression analysis was performed using limma in edgeR. *p*-values were adjusted for multiple testing using the Benjamini-Hochberg method. To plot the expression of differentially expressed hypoxia genes, we took all genes from the MSigDB Hallmark gene set for hypoxia and selected genes that were differentially expressed (Padj < 0.05) and had abs(log2 fold change) > 2 in all three comparisons 6 h vs. 0 h, 12 h vs. 0 h, and 24 h vs. 0 h. The heatmap was created using the package ComplexHeatmap and represents the log2expression mean-centered by row.

To generate a single volcano plot and functional enrichment analysis across all three comparisons (6 h vs. 0 h, 12 h vs. 0 h, and 24 h vs. 0 h), we computed summary statistics as follows. The *x*-axis of the volcano plot was computed as the average of the log2 fold change values across the three comparisons. The *y*-axis was computed as the average of the −log10(Padj) values across all three comparisons. To run Gene Set Enrichment Analysis (GSEA), we ranked each gene by its differential expression Padj value in each of the three comparisons and then computed the average rank across all three comparisons. The average rank was multiplied by the sign of the average fold change and then used to run pre-ranked GSEA with the MSigDB Hallmark gene sets.

### 2.4. Flow Cytometry

Cells were fixed by suspending in PBS with 4% PFA for 15 min and then suspended in Permeabilization Buffer (PBS, 0.5% Triton X-100) for 15 min, followed by ICC-Block (PBS, 5% NGS, 0.1% Triton X-100) for 30 min. Primary antibodies were diluted in ICC-Block, incubated for 1 h, washed and then incubated in secondary antibody for 30 min before final resuspension in PBS for analysis. All incubations were done in the dark, at room temperature, and on a rocker. Flow cytometry was performed using the Attune NxT Acoustic Focusing Cytometer (ThermoFisher, Waltham, MA, USA) and accompanying software. Antibodies used include GLUT1, GLUT3, and CA-9 at 1:1000 and PE Donkey anti-Rabbit (Biolegend, San Diego, CA, USA, 406421) at 1:500. Hoechst 33342 was used at 1:1500.

### 2.5. Immunofluroescence Microscopy

Cells were cultured in 8-well chamber slides (MatTek, Ashland, MA, USA, part no. CCS-8), washed in PBS, and fixed in ice-cold methanol for 12 min. After 5 min rehydration in ICC Wash (PBS, 0.1% Triton X-100), cells were rinsed in Permeabilization Buffer (PBS, 0.5% Triton X-100) for 15 min, blocked in ICC-Block (PBS, 5% NGS, 0.1% Triton X-100) for 30 min, and incubated with primary antibodies in ICC-Block for 1 h at room temperature. Next, cells were washed three times in ICC-Wash, incubated with secondary antibodies and Hoechst 33342 for 30 min at room temperature, then washed three times in ICC-Wash (5 min each), and mounted using ProLong Diamond Antifade Mountant (Thermo Fisher Scientific, Waltham, MA, USA, P36970). Antibodies used include GLUT1 (Atlas, Fort Collins, CO, USA, HPA058494), GLUT3 (Proteintech, Rosemont, IL, USA, 20403-I-AP), and CA-9 (Novus, Littleton, CO, USA, NB100-417SS) at 1:1000 dilution. Alexa Fluor 488 Goat anti-Rabbit (Invitrogen, Waltham, MA, USA, A11034) at 1:1500 and Hoechst 33342 (Invitrogen, Waltham, MA, USA, H3570) at 1:3000 dilution.

For xenograft tumor staining, FFPE blocks were sectioned at 5 µm thickness and mounted on slides. Slides were baked at 65 °C overnight and deparaffined by washing in xylene three times, for 7 min each, followed by passing through 100%, 70%, and 50% isopropanol and water for rehydration. Antigen retrieval was performed using an EnVision FLEX target retrieval DAKO solution high pH ((Agilent, Santa Clara, CA, USA, DM828) buffer and heated at 97 °C using a decloaking chamber for 20 min. Slides were washed in IHC-Wash (0.1 M Tris-HCl, 0.3 M NaCl, 0.1% Tween 20, and 7.7 mM NaN_3_, pH 7.6 at 25 °C) following by IHC-Block (5% normal goat serum, 0.1 M Tris-HCl, and 0.15 M NaCl, pH 7.6 at 25 °C) for 30 min at room temperature. Primary antibodies were diluted in IHC-Block and incubated at 4 °C overnight in a humidified chamber. Slides were then washed three times in IHC-Wash and incubated with a secondary antibody for 30 min at room temperature. Next, slides were washed three times in IHC-Wash, mounted using ProLong Diamond Antifade Mountant + DAPI (Thermo Fisher Scientific, Waltham, MA, USA, P36966), and stored overnight in the dark at room temperature to cure the mountant. Slides were imaged or stored at −20 °C for future analysis. Antibodies used include GLUT1 and CA-9 at 1:150 dilution, GLUT3 at 1:200 dilution, PIMO (Hydroxyprobe, Burlington, MA, USA, Hypoxyprobe1MAB1) at 1:100 dilution, E-Cadherin (Abcam, Cambridge, UK, ab76055) at 1:300 dilution, Alexa Flour 488 goat anti-mouse (Jackson Immunoresearch, West Grove, PA, USA, AB_2341099) and Rhodamine Red goat anti-rabbit (Jackson Immunoresearch, West Grove, PA, USA AB2338028) at 1:200 dilution.

Cells and tissue sections were imaged using a CSU-W1 SoRa Yokogawa spinning-disk field scanning confocal system assembled on a Nikon (Tokyo, Japan) ECLIPSE Ti2 inverted microscope with a Kinetix CMOS 10MP camera. For tissue, adjacent fields were imaged using a 4× objective (CFI60 Plan Apochromat λ 4×, NA 0.2, Nikon) or a 20× objective (CFI60 Plan Apochromat λ 20×, NA 0.75, Nikon), and then stitched together in Nikon NIS-Elements Microscope Imaging Software. Tissue culture cells were imaged with a 40× objective (CFI Plan Apochromat λ S 40XC Silicone, NA 1.25, Nikon) and maximum intensity projections of z-stacks were made in NIS-Elements. H&E samples were imaged using a Nikon BioPipeline Slide Scanner assembled on an Eclipse Ni-E microscope using a Color DS Fi3 camera and NIS-Elements Microscope Imaging Software.

### 2.6. Tumor Tissue Acquisition and PDX Passaging

Mouse xenograft and PDX tumors were conducted with Animal Care and Use Committee approval using the Experimental Mouse Shared Resource (University of Arizona Cancer Center). A total of 10 × 10^6^ cells (DU145, CTN1-2 or CT2-2) were injected subcutaneously into the rear flanks of male NOD/SCID/IL2Rγnull (NSG) mice (Jackson Laboratories), 2 injection sites per mouse. Tumor growth was measured 2 times/week and tumor volume estimated according to the formula: [(width)^2^ × length]/2. Mice were injected with pimonidazole (60 mg/kg by i.p.) 45 min prior to sacrifice to mark hypoxic tissue. Animals were terminated by cervical dislocation when total tumor reached 2000 mm^3^. PDX models were provided by Dr. Nora Navone (MD Anderson Cancer Center) [53]. PDX tumor specimens were propagated by serial passage into male SCID mice by subcutaneous flank injection. Tumors up to 1.5 cm in the longest measurable dimension were harvested and implanted into new host mice. The time that tumors take to grow to that size varies from approximately 3 to 12 weeks depending on the specific prostate cancer xenograft being used.

### 2.7. Image Analysis

Stitched images were cropped to 5 mm × 5 mm squares prior to analysis (equivalent to 2 × 2 stitched 4× images). Analysis was performed in NIS Element’s GA3 pipeline builder. All thresholds were determined empirically. For pimonidazole-injected xenografts, a minimum intensity threshold for the PIMO stain was determined and applied to each serial section to maintain consistency. First, a binary mask representing the tumor section was thresholded using the DAPI channel and eroded by 10 pixels to avoid sampling within tumor edges and all subsequent masks were restricted to regions that fall within this mask. Next, a binary mask representing autofluorescent red blood cells was thresholded in the DAPI channel. Finally, only regions containing high intensity of our stain of interest that were >30 µm in size were used for quantitation to avoid spurious bright pixels. The red blood cell mask was then subtracted from the PIMO mask. The PIMO mask was then dilated by 15 µm twice and the inner sections were removed to create the surrounding “donuts”. To generate a mask of the tumor background, the mask representing 30 µm away from the PIMO stain was excluded from the larger tumor mask. Finally, the mean fluorescence intensity of our marker of interest was taken for each region. Fold change was determined relative to the tumor background mean intensity.

Thresholds for patient-derived xenografts were determined as above except the tumor mask was determined using E-cadherin instead of DAPI and the marker binary was not dilated. Furthermore, the binary representing the marker of interest was determined by the triple stain and then kept consistent throughout the other two images. Total area of the stain of interest was then divided by the total tumor area and multiplied by 100 to determine the percentage of positive area.

## 3. Results

### 3.1. GLUT3 Is Highly Expressed in Hypoxic Immortalized Prostate Epithelial Cells

Hypoxia induces the expression of approximately 1% of the genome and, although there appears to be some common upregulated genes, the transcriptional response can vary dramatically in different cell types [51,54]. Since hypoxia is an early event in primary prostate tumor formation [55,56], we focused on understanding the hypoxic transcriptional response in non-tumorigenic prostate epithelial RWPE-1 cells with the goal of identifying candidate biomarkers that may identify hypoxia in normal, precancerous, and tumor tissue [57]. Furthermore, transcriptional responses change as cells spend more time in hypoxia. Therefore, we sought to find commonly upregulated genes throughout a hypoxic time course.

RNA was isolated and sequenced from cells cultured in either normoxic conditions (20% O_2_) or after shifting to hypoxia (1% O_2_) for 6, 12, and 24 h. Principle component analysis (PCA) revealed tight clustering of the normoxic samples away from all hypoxic samples, showing that the hypoxia response explains most of the variability in the dataset (Figure 1a). Additionally, consistent upregulation of MSigDB Hallmark hypoxia genes in each sample confirmed a common hypoxia response throughout the time course (Figure 1b). However, PCA did show progressive changes to the transcriptional response as cells experienced prolonged hypoxia treatment (Figure 1a). Therefore, to find potential biomarkers that persist throughout the hypoxia response, differentially expressed genes (DEGs) were identified throughout the time course (Appendix A). A volcano plot of averaged fold change and *p*-value statistics revealed a large subset of genes that were significantly upregulated throughout the time course (Figure 1c, Appendix A). Moreover, Gene Set Enrichment Analysis (GSEA) using MSigDB Hallmark gene sets identified hypoxia as the second highest ranked pathway activated across each time point (Figure 1d), demonstrating the strong hypoxia-induced transcriptional response in these cells. Myc Targets V1 and V2 also showed high normalized enrichment scores (NES), which is intriguing since HIFs are known to repress Myc target genes [58,59,60]. However, the interaction between HIF and Myc is complex, as they also cooperate to drive cancer cell growth and progression [61,62]. Alternatively, the upregulation of Myc target genes may simply reflect the immortalized state of RWPE1 cells [57].

Intriguingly, analysis of the DEGs also revealed that the SLC2A3 gene (encoding GLUT3) was markedly upregulated in all three hypoxia time points, and more significantly than either GLUT1 or CA9 (Figure 1b,c; Appendix A). Like GLUT1, GLUT3 is a class I member of the Na^2+^-independent family of facilitative glucose transporters and has a high affinity for glucose [39]. GLUT3 has widespread expression in tissues including the liver, kidney, and placenta, as well as neurons and parenchymal cells in the brain [39,41]. Importantly, GLUT3 is a member of the Hallmark hypoxia gene set and known to be upregulated in a variety of hypoxia models (Figure 1b). These include mouse blastocysts, rat and human neuronal lines, human embryonic stem cells, invasive ductal breast carcinoma, HeLa and choriocarcinoma cells as well as an ischemic brain injury mouse model of encephalopathy [63,64,65,66,67,68,69]. Moreover, GLUT3 expression in hypoxia is mediated by HIF-1α binding to a HRE in the GLUT3 promoter [65,67]. Notably, GLUT3 is also upregulated in luminal prostate epithelial PCT2 cells [70], but its expression in prostate cancer cells and its distribution in prostate tumors have not been described.

Lastly, we noticed that the SLC2A14 gene (encoding GLUT14) was upregulated in hypoxia at all time points and was nearly identical in significance to GLUT3 expression (Figure 1c; Appendix A). GLUT14 is a gene duplication of GLUT3, located 10 Mb upstream of GLUT3, and specifically expressed in the testis [71]. Since the GLUT14 cDNA shares ~95% identity with GLUT3 [71], we reasoned that RNA sequencing may have misidentified some GLUT3 messages as GLUT14. To test this, we performed RT-PCR on normoxic and hypoxic RWPE1 cells using validated primers sets that specifically distinguish between GLUT3 and GLUT14 cDNAs [71]. Whereas GLUT3 was expressed in cells cultured in either condition, we failed to detect any GLUT14 message in normoxic or hypoxic cells. Thus, our findings suggest that GLUT3 is specifically and significantly upregulated in response to hypoxia in cultured prostate epithelial cells.

### 3.2. GLUT3 Protein Levels Are Elevated in Hypoxic Prostate Epithelial Cells and Prostate Cancer Cells

We next determined how a hypoxia-mediated increase in GLUT3 expression compared to the established hypoxia biomarkers, CA9 and GLUT1, using flow cytometry. In this assay, we immunostained CA9, GLUT1, or GLUT3 in three different prostate/prostate cancer lines: immortalized RWPE1 cells, CTN-9 cells (a chemically transformed line derived from non-tumorigenic prostate epithelial PrEC-Hahn cells [52]), and DU145 (castrate-resistant brain metastatic tumor-derived prostate cancer line). Regardless of the cell type, immunofluorescence microscopy (IF) revealed an increase in expression of all three proteins in hypoxia, and that was enriched at the plasma membrane (Figure 2; Appendix A). Interestingly however, flow cytometry measurements of protein levels showed that the extent of upregulation varied depending on the cell type. For example, although GLUT1 and GLUT3 levels were nearly equivalent in hypoxic RWPE1 and CTN-9 cells, we observed a dramatic ~6–8 fold-increase in CA9 levels compared to cells grown in normoxic conditions (Figure 2; Appendix A). In hypoxic DU145 prostate cancer cell lines, GLUT3 showed the highest (albeit modest) increase in levels (2.6-fold) (Appendix A). Our results suggest that, similar to CA9 and GLUT1, GLUT3 protein levels are elevated in hypoxic prostate epithelial cells and cancer lines and may serve as a reliable biomarker of hypoxia in tissue.

### 3.3. GLUT3 Colocalizes with Pimonidazole-Labeled Regions of Hypoxia in Mouse Xenograft Tumors

To explore GLUT3′s potential as a biomarker of hypoxia in tumors, we next compared its distribution to CA9 and GLUT1 in mouse xenograft tumors labeled with pimonidazole. The extrinsic hypoxia marker pimonidazole is a 2-nitroimidazole-based molecule that is reduced in cells at low oxygen levels (<1.3%), forming stable adducts that can be detected immunohistochemically (e.g., Hypoyprobe) [72]. Indeed, pimonidazole-labeling is thought to specifically reveal acute hypoxia in vivo [73] and, in prostate cancer, associates with an upregulation of DNA repair, proliferation, and hypoxia-response genes [74]. In brief, male NSG mice were injected subcutaneously into the flank with either DU145 cells, or tumorigenic CTN1-2 or CTN2-2 cells (isolated from mouse xenograft PrEC-CTN9 tumors [72]). One hour prior to harvesting the tumors, mice were intravenously injected with pimonidazole to label pockets of hypoxia which, in our experience, are always present in large, solid xenograft tumors. Formalin-fixed paraffin-embedded (FFPE) tumors were then serially sectioned to 5 µm thickness and stained with hematoxylin and eosin (H&E) or immunostained for pimonidazole (PIMO) and either CA9, GLUT1, or GLUT3. Expectedly, numerous pockets of PIMO staining were present in all tumors, consisting of discrete gradients of irregular shapes and sizes (Figure 3 and Figure 4). PIMO labeling was commonly located near or surrounding necrotic patches but not necessarily defined by their distance from blood vessels identified in the H&E, likely explained by the faulty tumor vasculature. GLUT3 staining was not uniform throughout the tumor but, instead, formed discrete patches that were strikingly similar to PIMO at low magnification, and that frequently overlapped or juxtaposed the PIMO signal (Figure 3 and Figure 4, yellow arrows). Possibly, these neighboring patches of GLUT3 signal reveal regions of hypoxia that were currently above the oxygen threshold for PIMO detection. However, since PIMO only captures a snapshot of currently hypoxic tissue, the adjacent GLUT3 staining may highlight regions of chronic hypoxia within the tumor. At higher magnification, GLUT3 staining was clearly concentrated at the plasma membrane in positive cells, as expected for a glucose transporter (Figure 3 and Figure 4, boxed regions). CA9 staining appeared similar to GLUT3, whereas GLUT1 staining was modest in most tumors that we examined.

To quantify the degree of co-localization between PIMO and the three hypoxia markers, we acquired multiple images from two tumors generated from each cell line (a total of 6 tumors) and used automated image masking to define PIMO-positive regions (Figure 5a,b). Masks were then dilated to >0–15 µm and >15–30 µm to capture any non-overlapping, peripheral signals of the hypoxia markers (Figure 5a,b). Mean fluorescent intensities of CA9, GLUT1, or GLUT3 were then measured within these three zones and displayed as measurements of fold change (after dividing by the mean surrounding background signal delineated as >30 µm distal to the PIMO signal) (Figure 5c). Remarkably, GLUT3 appeared to outperform the other hypoxia markers in half of the tumors we examined (including both DU145 tumors), displaying the highest signal enrichment within the PIMO-positive zone (0 µm). In the other tumors (CTN1-2 #1, CTN2-2 #1 and #2), we found that CA9 signal was highest, primarily in the PIMO-positive zone. Thus, our findings suggest that GLUT3 is a biomarker of hypoxia in xenograft prostate tumors. These measurements also show that hypoxia markers frequently display a radial gradient in their spatial patterns, likely reflecting the oxygen gradients present in tumor tissue.

### 3.4. A Combination of Biomarker Stains Provides the Best Coverage of Hypoxic Regions in Patient-Derived Xenograft (PDX) Models of Prostate Cancer

Our findings indicate that GLUT3 can be a reliable biomarker of hypoxia in tumors but, by itself, does not identify all hypoxic/PIMO-positive regions, likely due to tumor heterogeneity. Therefore, we next examined the spatial distribution of GLUT3 in tumors compared to a combination of the classical biomarkers CA9/GLUT1 or a CA9/GLUT1/GLUT3 triple stain. (Our antibodies against CA9, GLUT1, and GLUT3 are all rabbit polyclonals and can easily be used in mixtures to reveal a cumulative pattern). For these experiments, we performed immunofluorescence using two patient-derived xenograft (PDX) models of prostate cancer (PCa) obtained from the Movember Global Action Plan 1: a primary prostate adenocarcinoma (MDA PCa 173-2, androgen naïve, Gleason score 3 + 4) and a bone metastatic (met) tumor (MDA PCa 183-A, Androgen Receptor positive) [53]. FFPE-tumors were serially sectioned to 5 µm thickness and stained with hematoxylin and eosin (H&E) or immunostained for E-Cadherin (to mark cell borders) and either GLUT3, GLUT1/CA9, or GLUT1/GLUT3/CA9. As we observed in the previous mouse xenograft tumors, GLUT3 staining alone in both the primary and bone metastatic PDX tumors was remarkable, appearing as distinct patches and stripes with a graded distribution and enriched at the plasma membrane in individual cells (Figure 6 and Figure 7). A qualitative examination of the various stains showed that the triple stain (GLUT1/GLUT3/CA9) was very similar to GLUT3 alone (Appendix A). However, the signal in some triple-stained regions appeared more intense than in the GLUT3 stain alone. Moreover, new regions of signal were identified in the serial sections with the triple stain that were not seen in the GLUT3 stained sections.

To quantify the efficacy of the hypoxia stains in the PDX tumor sections, we again used automated image masking to identify their patterns within multiple stitched tumor images, and then measured the percent area they covered within the total image. An example of our image masking is shown in Figure 8a,b. In this analysis, we measured staining within serial sections of two different bone-derived PDX tumors and one primary PDX tumors. We found that within one of the bone-derived tumors (Tumor #2), the area covered by the triple stain was comparable to that measured for GLUT3 alone (Figure 8c), suggesting that most (if not all) signal emanating from the triple staining was due to GLUT3. However, in the other met and primary PDX sections, signal from the triple stain clearly encompassed a greater overall tumor area than could be explained by only GLUT3 (Figure 8c). For example, in the primary PDX sample, the area of tumor that the GLUT3 staining occupied was less than half that of the triple stain (Figure 8c). In any case, our measurements indicate that the addition of anti-GLUT3 antibody increased the area of hypoxia detection by 2.6–22.2 fold. Thus, our findings suggest that GLUT3 labels hypoxic regions in prostate cancer PDX models and should be represented within a panel of other hypoxia biomarkers.

## 4. Discussion

Hypoxia is a common feature of the tumor microenvironment and is particularly problematic in prostate cancer because it causes resistance to radiotherapy, promotes metastasis, and predicts a poor prognosis in patients [44,75,76]. In our study, we used RNA-seq to screen for genes that were upregulated in hypoxic prostate epithelial cells with the ultimate goal of identifying new endogenous biomarkers of hypoxia for diagnostic use in prostate cancer. This bioinformatic approach identified the glucose transporter, GLUT3, as a significantly upregulated gene in immortalized RWPE1 cells. The discovery of hypoxia-mediated upregulation of GLUT3 is not new though; GLUT3 is upregulated in a variety of hypoxic cells including osteosarcoma, invasive ductal breast carcinoma, and choriocarcinoma cells [63,64,65,66,67,68,69,70]. Furthermore, the Human Protein Atlas lists high levels of GLUT3 expression as prognostically unfavorable in renal, stomach, and colorectal cancers [77,78]. GLUT3 was also identified as part of a 28 gene hypoxia-related prognostic signature for localized prostate cancer [70]. High signature scores associated with poor recurrence free survival for prostatectomy-treated patients and predicted both recurrence in patients receiving radiotherapy as well as metastasis. Notably, CA9 and GLUT1 were not among the genes in this signature, but GLUT14 was [70]. Considering the high degree of identity between the GLUT3 and GLUT14 genes, it is likely that GLUT3 transcripts were mistaken as GLUT14, as in our current study. In any case, a hypoxia-induced switch to GLUT3 as the preferred glucose transporter is consistent with the observations that GLUT3 is found in neurons and other cell types with high energy demands and has greater affinity for glucose compared to GLUT1 [79,80].

Although we first identified GLUT3 as a significantly upregulated DEG in hypoxic immortalized prostate epithelial cells, we showed that GLUT3 was also upregulated in response to hypoxia in other cell lines and models including transformed PrEC-Hahn cells, brain metastatic tumor-derived DU145 cells, xenograft tumors formed from these cell lines, and two different prostate cancer PDX specimens (isolated from a primary tumor and a bone metastatic tumor). This suggests that GLUT3 may serve as a full spectrum biomarker of hypoxia in prostate cancer from precancerous cells through progression to malignancy. The GLUT3 staining we observed in tumor tissue showed discrete patterns and gradients, concentrated with pimonidazole, and covered the greatest area within tumors as compared to the hypoxia markers CA9 and GLUT1. Although GLUT3 seems a preferred biomarker to identify hypoxia in vivo, this was not always the case given the genomic and microenvironmental heterogeneity in tumors. We agree with the recommendation that, instead of focusing on individual genes, we should employ tools that recognize a panel of hypoxia genes and markers to accurately reveal all tumoral hypoxia [51]. To that end, we used a cocktail of antibodies against GLUT1, GLUT3, and CA9 (the triple stain) which, by our measurements, was the most informative approach. Future studies should focus on how the triple hypoxia stain performs on biopsy specimens from prostate cancer patients as well as its utility in other hormone-related cancers such as breast, ovary, and testis.

## Figures and Tables

**Figure 1 diagnostics-12-00676-f001:**
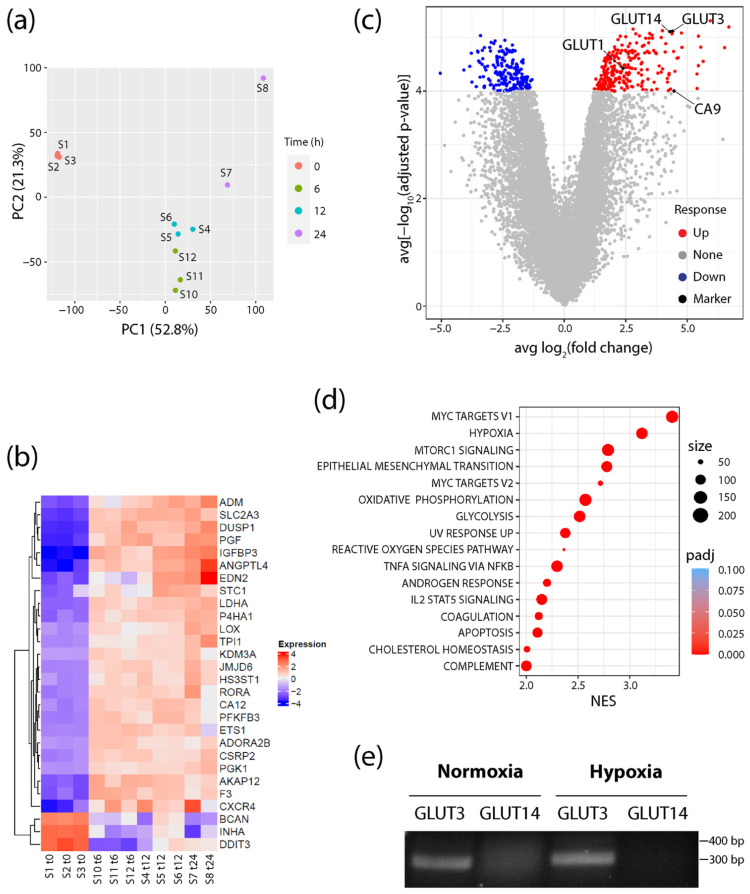
GLUT3/SLC2A3 is highly upregulated in hypoxic conditions. (**a**) Principal component analysis plot showing RNA-seq profiles of prostate cells samples (S) at 0, 6, 12, and 24 h after exposure to hypoxic conditions (1% O_2_). (**b**) Heatmap represents the log2 (expression) values of all genes that are differentially expressed (Padj < 0.05 and abs [log2FC] > 2) in all three time points relative to the 0 h normoxia samples (S1–3), and that are also members of the MSigDB Hallmark hypoxia gene set. Each row has been mean-centered to better show changes in expression over time. (**c**) Volcano plot showing the average log2FC over each time point (6, 12, and 24 h) relative to 0 h normoxia samples (*x*-axis), and average significance (computed as −log10Padj) over each time point relative to 0 h normoxia samples (*y*-axis). (**d**) Dot plot shows all upregulated MSigDB Hallmark gene sets with Padj < 0.05 and normalized enrichment score (NES) > 2 when performing gene set enrichment analysis on the average rank times the sign of the fold change across each of the three time points (6, 12, and 24 h) relative to 0 h normoxia samples. The size of the dot indicates the total number of genes annotated to the Hallmark gene set. Color of the dot indicates the Padj value. (**e**) Primers specific for GLUT3 or GLUT14 were used for RT-PCR to assess expression in total RNAs isolated from normoxic and 24 h hypoxic RWPE1 cells. Predicted sizes of the PCR products: GLUT3, 299 bp; GLUT14, 370 bp. Note, this assay was used to simply detect the transcript and was not a quantitative assay of transcript levels.

**Figure 2 diagnostics-12-00676-f002:**
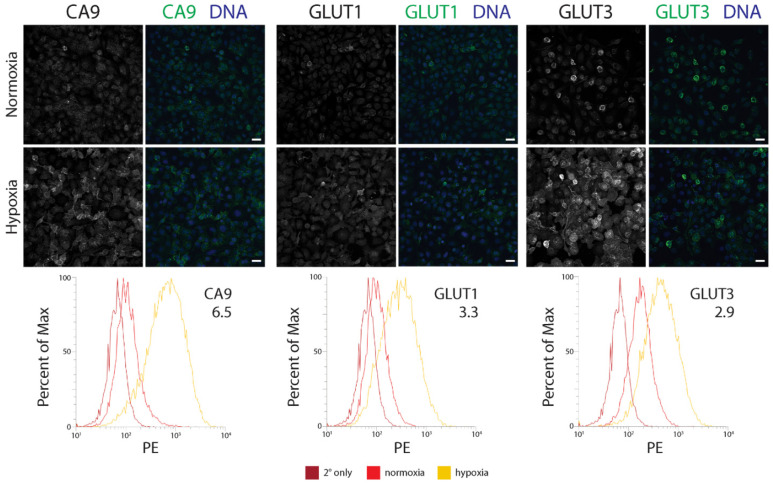
The GLUT3 protein is upregulated in hypoxic immortalized prostate epithelial cells. (**Upper** and **Middle** panels) RWPE1 cells were cultured in either normoxic or under chronic hypoxic (1% O_2_) conditions for 48 h and then processed for immunofluorescence microscopy. Cells were stained for either CA9, GLUT1, or GLUT3 (green, **Upper** and **Middle** panels). DNA, blue. Scale, 25 µm. (**Lower** panels) Comparison of flow cytometry profiles of CA9, GLUT1, or GLUT3 in cells cultured in normoxia (red) versus hypoxia (yellow). Cells stained with only secondary antibody is shown (dark red). The fold increase in level of the indicated proteins is denoted. PE, phycoerythrin.

**Figure 3 diagnostics-12-00676-f003:**
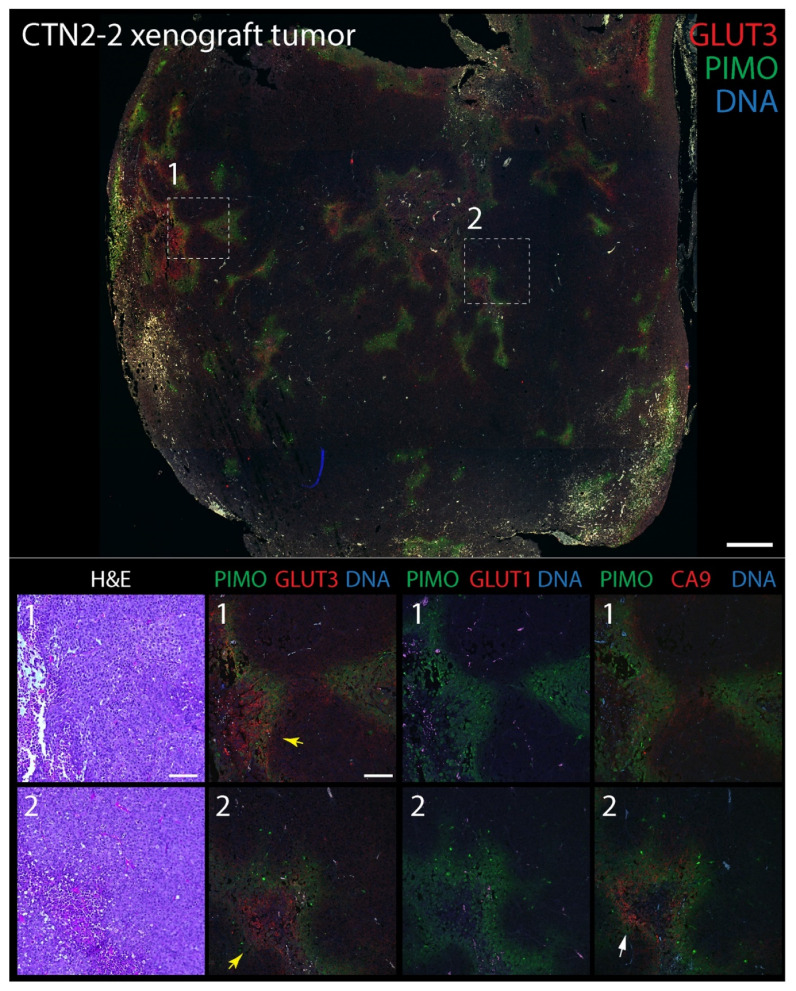
GLUT3 co-localizes with pimonidazole (Hypoxyprobe) in mouse xenograft tumors formed from prostate epithelial CTN-2-2 cells. Non-tumorigenic immortalized prostate epithelial PrEC-Hahn cells were transformed by transiently inducing chromosomal instability. The resulting tumorigenic line was named CTN-9, which produced malignant solid xenograft tumors in subcutaneously injected NSG mice [52]. Cell lines isolated from these tumors were named CTN1-2 and 2-2 lines [52]. (**Upper** panel) Single tumor section stained for pimonidazole (PIMO) (green) and GLUT3 (red). DNA, blue. Boxed regions are shown at higher magnification in the lower panels. (**Lower** panels) Higher magnification images from boxed regions 1 or 2 in upper panel. Serial tissue sections were stained for hematoxylin and eosin (H&E), PIMO (green), and GLUT3, GLUT1 or CA9 (red) as indicated. Concentrations of GLUT3 (yellow arrows) and CA9 (white arrows) that juxtapose with PIMO are marked. Dashed white boxes show GLUT3 concentrated at the plasma membrane in tumor cells (insets). Scale, 100 µm.

**Figure 4 diagnostics-12-00676-f004:**
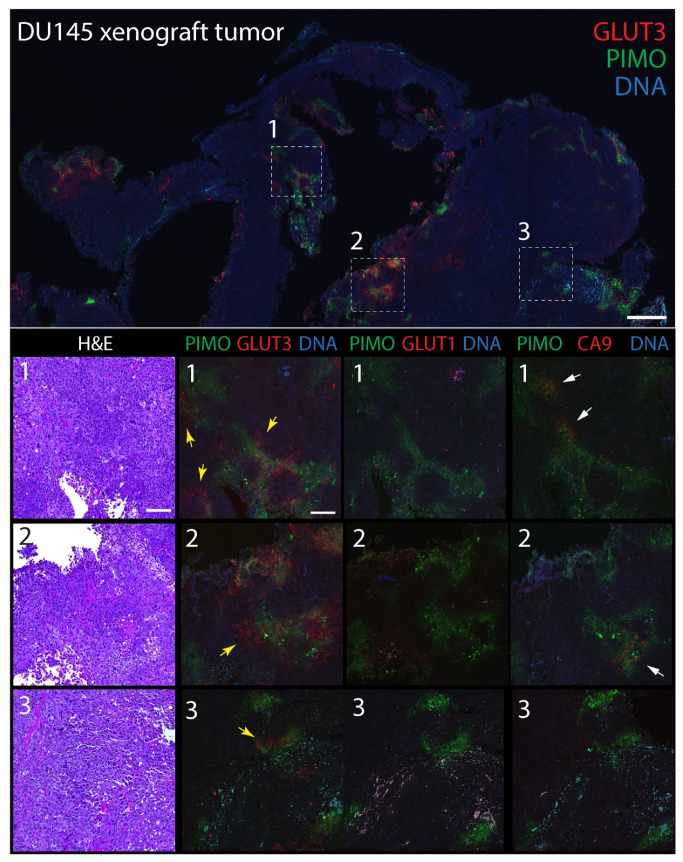
GLUT3 co-localizes with pimonidazole (Hypoxyprobe) in mouse xenograft tumors formed from prostate cancer DU145 cells. (**Upper** panel) Single tumor section stained for pimonidazole (PIMO) (green) and GLUT3 (red). DNA, blue. Boxed regions are shown at higher magnification in the lower panels. (**Lower** panels) Higher magnification images from boxed regions 1, 2, or 3 in upper panel. Serial tissue sections were stained for hematoxylin and eosin (H&E), PIMO (green), and GLUT3, GLUT1 or CA9 (red) as indicated. Concentrations of GLUT3 (yellow arrows) and CA9 (white arrows) that juxtapose with PIMO are marked. Dashed white boxes show GLUT3 concentrated at the plasma membrane in tumor cells (insets). Scale, 100 µm.

**Figure 5 diagnostics-12-00676-f005:**
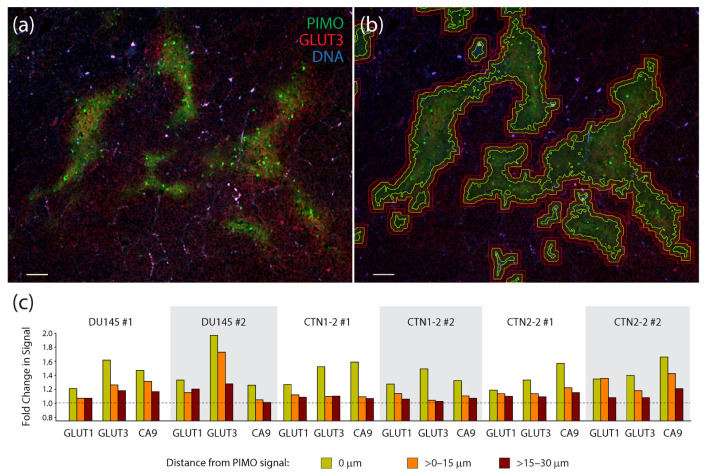
Measurements of the co-localization of hypoxia markers in xenograft tumors show a high degree of association between pimonidazole and GLUT3. (**a**) Single section from a CTN2-2 tumor immunostained for pimonidazole (PIMO) (green) and GLUT3 (red). DNA, blue. Scale, 100 µm. (**b**) To quantify co-localization of hypoxia markers GLUT1, GLUT3, and CA9 with PIMO, binary tracings were placed around the PIMO signal (green lines) defined a threshold of intensity specific to each tumor. Binaries were then dilated >0–15 µm (orange lines) and >15–30 µm (red lines) to capture the adjacent signal of each hypoxia marker. Same image as in (**a**). (**c**) Graphs show fold change in mean fluorescence intensities of hypoxia markers GLUT1, GLUT3 and CA9 within three PIMO-positive or adjacent regions captured within the expanded binary masks: 0 µm where the PIMO signal resides, >0–15 µm (orange bars) and >15–30 µm (red bars). Mean fluorescence intensities within these three regions was divided by ‘background’ signal from regions >30 µm from the PIMO signal. A dashed line indicates no change in signal relative to the background. Four optical fields in tissue sections from two different DU145, CTN1-2 and CTN2-2 tumors were analyzed (six different tumors examined in total).

**Figure 6 diagnostics-12-00676-f006:**
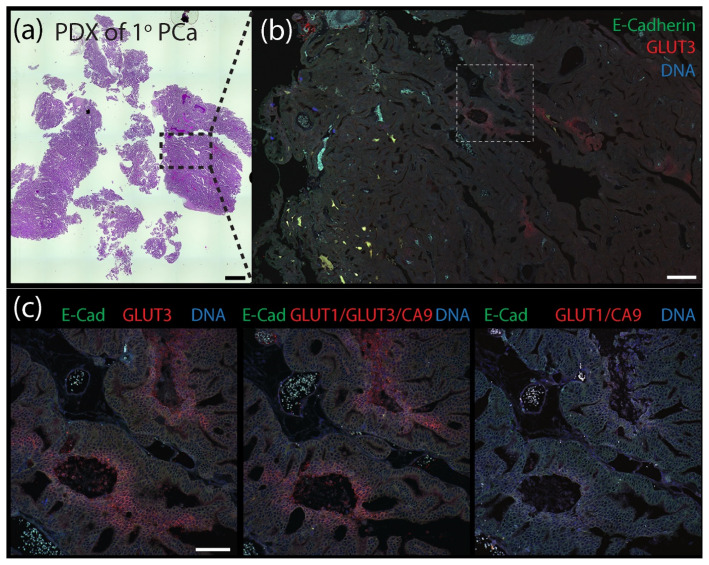
Patient-derived xenograph (PDX) model of primary prostate cancer contains distinct pockets of GLUT3 staining. (**a**) Patient-derived xenograft (PDX) from a primary prostate tumor stained with H&E. Scale, 1.5 mm. (**b**) Section of PDX tumor immunostained for E-Cadherin (green) to mark cell borders and GLUT3 (red). DNA, blue. Boxed region is shown at higher magnification in (**c**). Scale, 250 µm. (**c**) Serial tissue sections were immunostained for either GLUT3 (red, left panel), a cocktail containing anti-GLUT1 and anti-CA9 antibodes (red, right panel), or antibodies against all three proteins GLUT1, GLUT3, and CA9 (red, middle panel) as indicated. Dashed white boxes show GLUT3 concentrated at the plasma membrane in tumor cells (insets). Scale, 100 µm.

**Figure 7 diagnostics-12-00676-f007:**
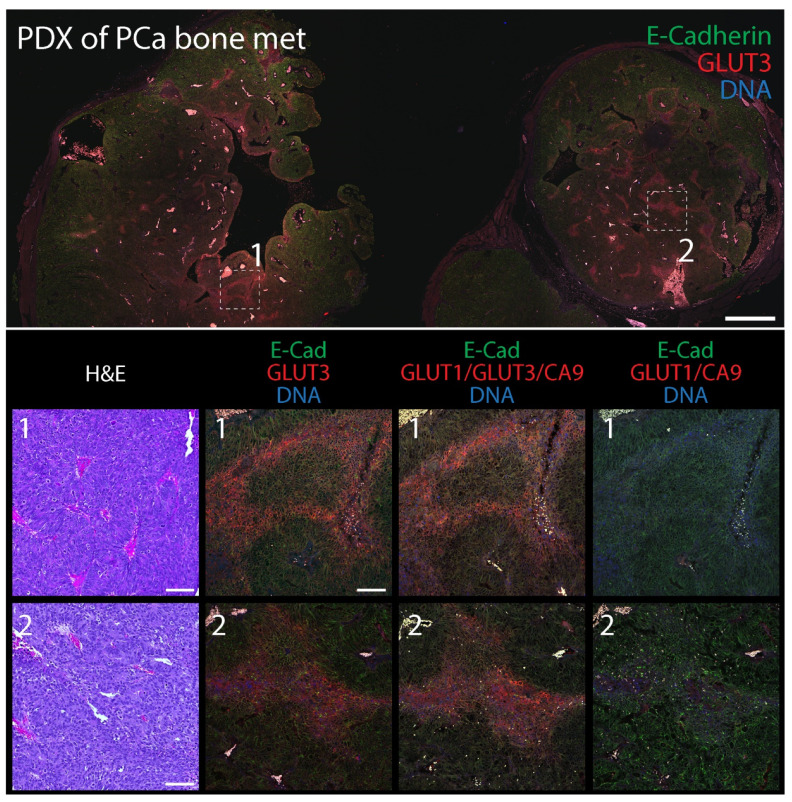
Patient-derived xenograph (PDX) model of bone metastatic prostate cancer contains distinct pockets of GLUT3 staining. (**Upper** panel) Section of PDX tumor immunostained for E-Cadherin (green) to mark cell borders and GLUT3 (red). DNA, blue. Boxed regions are shown at higher magnification (**Lower** panels). Scale, 500 µm. (**Lower** panels) Higher magnification images from boxed regions 1 or 2 in upper panel. Serial tissue sections were stained with H&E (column 1) or immunostained for either GLUT3 (red, column 2), a cocktail containing anti-GLUT1, GLUT3, and CA9, antibodies (red, column 3), or antibodies against GLUT1 and CA9 (red, column 4) as indicated. Dashed white boxes show GLUT3 concentrated at the plasma membrane in tumor cells (insets). Scale, 100 µm.

**Figure 8 diagnostics-12-00676-f008:**
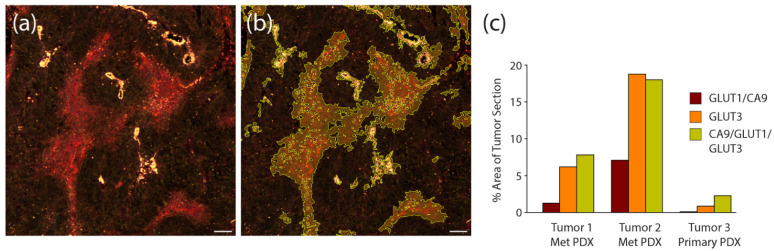
Compared to hypoxia biomarkers GLUT1 and CA9, GLUT3 labels larger areas within PDX tumor sections. (**a**) Patient-derived xenograft (PDX) from a primary prostate tumor immunostained with the triple anti-CA9, GLUT1, and GLUT3 antibody cocktail. Scale, 100 µm. (**b**) Same image as in (**a**) with binary mask to outline regions of the triple stain. (**c**) Graph shows measurements of total area of each stain within 3 different PDX prostate tumors. Three sequential sections were obtained from each tumor, stained for either CA9/GLUT1, GLUT3, or the triple stain CA9/GLUT1/GLUT3, and imaged. Total area of staining was quantified by masking regions of high intensity staining using Nikon Elements image analysis software. Note, in the metastatic prostate PDX tumor samples, the vast majority of the signal observed in the triple stain is due to the GLUT3 stain. However, in the primary prostate PDX, GLUT3 staining alone is not as effective at revealing putative hypoxic regions compared to the triple stain.

## Data Availability

The data supporting the findings of this study are available from the corresponding authors (Megha Padi, Anne Cress and Gregory Rogers) upon reasonable request. RNAseq data files are available from the Gene Expression Omnibus (GEO) repository (GSE196634).

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
