# Peer review of "GLUT3/SLC2A3 Is an Endogenous Marker of Hypoxia in Prostate Cancer Cell Lines and Patient-Derived Xenograft Tumors"

_diagnostics, 2022, doi:10.3390/diagnostics12030676_

Round 1

Reviewer 1 Report

Recognition of hypoxia tumour and cells can significantly importance in the design of therapeutic regiments. Authors showed that GLUT3 expression is strong marker of hypoxia phenotype. Therefore I would like to recommend acceptance of this manuscript, only minor points can be taken.

Double space - lines 49, 51, 66, 185, 186, 272

Author Response

Letter to the Reviewers

We thank the reviewers for their thoughtful comments.   We address each of the reviewers’ concerns below point-by-point.  Reviewer comments are in italics.

Reviewer 1

Recognition of hypoxia tumour and cells can significantly importance in the design of therapeutic regiments. Authors showed that GLUT3 expression is strong marker of hypoxia phenotype. Therefore I would like to recommend acceptance of this manuscript, only minor points can be taken.

Double space - lines 49, 51, 66, 185, 186, 272

Apparently, several errors appeared in our manuscript when it was converted to Diagnostic’s journal format.  We have corrected the double spaces (and other errors) in the revision.

Reviewer 2 Report

Ryniawec et al. reports over the role of GLUT3 as biomarker for tumor progression, specifically focusing on cancer of the prostate. They propose that GLUT3 maybe used as hypoxia marker in prostate cancer, by using cancer cell lines, xenografts and PDX-derived models. The study is clearly written and presented, data are supportive of the conclusions made. I have no major remarks.

Minor: microscopic images need to be shown also at higher magnification to better appreciate the signal localization.

Does GLUT3 colocalize with other known hypoxic markers?

Since hypoxia is a common feature of tumor progression , is GLUT3 relevant for other cancer types?

Author Response

Letter to the Reviewers

We thank the reviewers for their thoughtful comments.   We address each of the reviewers’ concerns below point-by-point.  Reviewer comments are in italics.

Reviewer 2

Ryniawec et al. reports over the role of GLUT3 as biomarker for tumor progression, specifically focusing on cancer of the prostate. They propose that GLUT3 maybe used as hypoxia marker in prostate cancer, by using cancer cell lines, xenografts and PDX-derived models. The study is clearly written and presented, data are supportive of the conclusions made. I have no major remarks.

Minor: microscopic images need to be shown also at higher magnification to better appreciate the signal localization.

As suggested, we have inserted high magnification insets of the GLUT3 immunostaining for all of the xenograft tumors (new Figures 3, 4, 6, and 7).

Does GLUT3 colocalize with other known hypoxic markers?

In our manuscript, we show that GLUT3 co-localizes with pimonidazole, an extrinsic molecule that specifically labels regions of acute hypoxia in tissue and tumors.  Pimonidazole is the gold standard hypoxia label in the field.  When comparing the localization of GLUT3 with two other hypoxia markers, CA9 and GLUT1, our results show that no two hypoxia markers behave the same and frequently do not show much overlap in xenograft tumors (highlighting the heterogenous nature of tumors).  We chose CA9 and GLUT1 because these two proteins are accepted as the premier hypoxia biomarkers in prostate cancer.

Since hypoxia is a common feature of tumor progression , is GLUT3 relevant for other cancer types?

We have added the following sentence in the Discussion section and added two additional references: “Furthermore, the Human Protein Atlas lists high levels of GLUT3 expression as prognostically unfavorable in renal, stomach, and colorectal cancers [78, 79].”